# Threshold Bandit, With and Without Censored Feedback

**Jacob Abernethy**
Department of Computer Science
University of Michigan
Ann Arbor, MI 48109
jabernet@umich.edu

**Kareem Amin**
Department of Computer Science
University of Michigan
Ann Arbor, MI 48109
amkareem@umich.edu

**Ruihao Zhu**
AeroAstro&CSAIL
MIT
Cambridge, MA 02139
rzhu@mit.edu

## Abstract

We consider the *Threshold Bandit* setting, a variant of the classical multi-armed bandit problem in which the reward on each round depends on a piece of side information known as a *threshold value*. The learner selects one of $K$ actions (arms), this action generates a random sample from a fixed distribution, and the action then receives a unit payoff in the event that this sample exceeds the threshold value. We consider two versions of this problem, the *uncensored* and *censored* case, that determine whether the sample is always observed or only when the threshold is not met. Using new tools to understand the popular UCB algorithm, we show that the uncensored case is essentially no more difficult than the classical multi-armed bandit setting. Finally we show that the censored case exhibits more challenges, but we give guarantees in the event that the sequence of threshold values is generated optimistically.

## 1 Introduction

The classical *Multi-armed Bandit* (MAB) problem provides a framework to reason about sequential decision settings, but specifically where the learner's chosen decision is intimately tied to information content received as feedback. MAB problems have generated much interest in the Machine Learning research literature in recent years, particularly as a result of the changing nature in which learning and estimation algorithms are employed in practice. More and more we encounter scenarios in which the procedure used to make and exploit algorithmic predictions is exactly the same procedure used to capture new data to improve prediction performance. In other words it is increasingly harder to view *training* and *testing* as distinct entities.

MAB problems generally involve repeatedly making a choice between one of a finite (or even infinite) set of actions, and these actions have historically been referred to as *arms* of the bandit. If we "pull" arm $i$ at round $t$, then we receive a reward $R_i^t \in [0, 1]$ which is frequently assumed to be a stochastic quantity that is drawn according to distribution $D_i$. Typically we assume that $D_i$ are heterogeneous across the arms $i$, whereas we assume the samples $\{R_i^t\}_{t=1,...,T}$ are independently and identically distributed according to the fixed $D_i$ across all times $t$.[1] Of course, were the learner to have full knowledge of the distributions $D_i$ from the outset, she would presumably choose to pull the arm whose expected reward $\mu_i$ is highest. With that in mind, we tend to consider the (expected) *regret* of the learner, defined to be the (expected) reward of the best arm minus the (expected) reward of the actual arms selected by the learner.

Early work on MAB problems (Robbins, 1952; Lai and Robbins, 1985; Gittins et al., 2011) tended to be more focused on asymptotic guarantees, whereas more recent work (Auer et al., 2002; Auer, 2003)

has been directed towards a more "finite time" approach in which we can bound regret for fixed time horizons $T$. One of the best-known and well-studied techniques is known as the *Upper Confidence Bound* (UCB) algorithm (Auer et al., 2002; Auer and Ortner, 2010). The magic of UCB relies on a very intuitive policy framework, that a learner should select decisions by maximizing over rewards estimated from previous data but *only after* biasing each estimate according to its uncertainty. Simply put, one should choose the arm that maximizes the "mean plus confidence interval," hence the name Upper Confidence Bound.

In the present paper we focus on the *Threshold Bandit* setting, described as follows. On each round $t$, a piece of side information is given to the learner in the form of a real number $c^t$, the learner must then choose arm $i$ out of $K$ arms, and this arm produces a value $X_i^t$ drawn from a survival distribution with survival function $F_i(x) = \Pr(X_i^t \geq x)$. The reward to the learner is not $X_i^t$ itself but is instead the binary value $R_i^t = \mathbb{I}[X_i^t \geq c^t]$; that is, we receive a unit reward when the sample $X_i^t$ exceeds the threshold value $c^t$, and otherwise we receive no reward. For a fixed value of $c^t$, each arm $i$ has expected payoff $\mathbb{E}[R_i^t] = F_i(c^t)$. Notice, crucially, that the arm with the greatest expected payoff *can vary significantly* across different threshold values.

This abstract model has a number of very natural applications:

1. **Packet Delivery with Deadlines:** FedEx receives a stream of packages that need to be shipped from source to destination, and each package is supplied with a *delivery deadline*. The goal of the FedEx routing system is to select a transportation route (via air or road or ship, etc.) that has the highest probability of on-time arrival. Of course some transportation schemes are often faster (e.g. air travel) but have higher volatility (e.g. due to poor weather).
2. **Supplier Selection:** Customers approach a manufacturing firm to produce a product with specific quality demands. The firm must approach one of several suppliers to contract out the work, but the firm is uncertain as to the capabilities and variabilities of the products each supplier produces.
3. **Dark Pool Brokerage:** A financial brokerage firm is asked to buy or sell various sized bundles of shares, and the brokerage aims to offload the transactions onto one of many *dark pools*, i.e. financial exchanges that match buyers and sellers in a confidential manner (Ganchev et al., 2010; Amin et al., 2012; Agarwal et al., 2010). A standard dark pool mechanism will simply execute the transaction if there is suitable liquidity, or will reject the transaction when no match is made. Of course the brokerage gets paid on commission, and simply wants to choose the pool that has the highest probability of completion.

What distinguishes the Threshold Bandit problem from the standard stochastic multi-armed bandit setting are two main features:

1. The regret of the learner will be measured in comparison to the best *policy* rather than to simply the best *arm*. Note that the optimal offline policy may incorporate the threshold value $c^t$ before selecting an arm $I^t$.
2. Whereas the standard stochastic bandit setting assumes that we observe the reward $R_{I^t}^t$ of the chosen arm $I^t$, in the Threshold Bandit setting we consider two types of feedback.
    (a) **Uncensored Feedback:** After playing arm $I^t$, the learner observes the sample $X_{I^t}^t$ regardless of the threshold value $c^t$. This is a natural model for the FedEx routing problem above, wherein one learns the travel time of a package regardless of the deadline having been met.
    (b) **Censored Feedback:** After playing $I^t$, the learner observes a null value when $X_{I^t}^t \geq c^t$, and otherwise observes $X_{I^t}^t$. This is a natural model for the Supplier Selection problem above, as we would only learn the product's quality value when the customer rejects what is received from the supplier.

In the present paper we present roughly three primary results. First, we provide a new perspective on the classical UCB algorithm, giving an alternative proof that relies on an interesting potential function argument; we believe this technique may be of independent interest. Second, we analyze the Threshold Bandit setting when given uncensored feedback, and we give a novel algorithm called DKWUCB based on the Dvoretzky-Kiefer-Wolfowitz inequality (Dvoretzky et al., 1956). We show, somewhat surprisingly, that with uncensored feedback the regret bound is no worse than the standard

stochastic MAB setting, suggesting that despite the much richer policy class one has nearly the same learning challenge. Finally, we consider learning in the censored feedback setting, and propose an algorithm KMUCB, akin to the *Kaplan-Meier estimator* (Kaplan and Meier, 1958). Learning with censored feedback is indeed more difficult, as the performance can depend significantly on the *order* of the threshold values. In the worst case, when threshold values are chosen in an adversarial order, the cost of learning scales with the number of unique threshold values, whereas one can perform significantly better under certain constraints on optimistic assumptions on the order or even a random order.

## 2 A New Perspective on UCB

Before focusing on the Threshold Bandit problem, let us turn our attention to the classical stochastic MAB setting and give another look at the UCB algorithm. We will now provide a modified proof of the regret bound of UCB that relies on a potential function. Potential arguments have proved quite popular in studying adversarial bandit problems (Auer et al., 2003; Audibert and Bubeck, 2009; Abernethy et al., 2012; Neu and Bartók, 2013; Abernethy et al., 2015), but have received less use in the stochastic setting. This potential trick is the basis for forthcoming results on the Threshold Bandit.

Let $D_i$ be a distribution on the reward $R_i^t$, with support on $[0, 1]$. We imagine the rewards $R_i^1, \ldots, R_i^T \overset{\text{i.i.d.}}{\sim} D_i$, whose mean $\mathbb{E}[R_i^t] = \mu_i$. A *bandit algorithm* is simply a procedure that chooses a random arm/action $I^t$ on round $t$ as a function of the set of past observed (action, reward) pairs, $(I^1, R_{I^1}^1), \ldots, (I^{t-1}, R_{I^{t-1}}^{t-1})$. Finally, let $N_i^t := \sum_{\tau=1}^{t-1} \mathbb{I}[I^\tau = i]$ and define the empirical mean estimator at time $t$ to be $\hat{\mu}_i^t := \frac{\sum_{\tau=1}^{t-1} \mathbb{I}[I^\tau = i] R_{I^\tau}^\tau}{N_i^t}$.

We assume we are given a particular *deviation bound* which provides the following guarantee,

$$\Pr\left(|\mu_i - \hat{\mu}_i^t| > \varepsilon \mid N_i^t \geq N\right) \leq f(N, \varepsilon),$$

where $f(\cdot)$ is some function, continuous in $\varepsilon$ and monotonically decreasing in both parameters, that controls the probability of a large deviation. While UCB relies specifically on the *Hoeffding-Azuma inequality* (Cesa-Bianchi and Lugosi, 2006), for now we leave the deviation bound in generic form. This will be useful in following sections.

Given $f(\cdot, \cdot)$, what is of interest to our present work is a pair of functions that allow us to convert between values of $\varepsilon$ and $N$ in order to guarantee that $f(N, \varepsilon) \leq \delta$ for a given $\delta > 0$. To this end define

$$
\begin{aligned}
\sharp(\varepsilon, \delta) &:= \min\{N : f(N, \varepsilon/2) \leq \delta\}, \\
\varsigma(N, \delta) &:= \begin{cases} \inf\{\varepsilon : f(N, \varepsilon) \leq \delta\} & \text{if } N > 0; \\ 1 & \text{otherwise,} \end{cases}
\end{aligned}
$$

We will often omit the $\delta$ in the argument to $\sharp(\cdot), \varsigma(\cdot)$. Note the key property that $\varsigma(N, \delta) \leq \varepsilon/2$ for any $N \geq \sharp(\varepsilon, \delta)$.

We can now define our variant of the UCB algorithm for a fixed choice of $\delta > 0$.

$$\textbf{UCB Algorithm:} \qquad \text{on round } t \text{ play } I^t = \underset{i}{\arg\max}\left\{\hat{\mu}_i^t + \varsigma(N_i^t, \delta)\right\} \tag{1}$$

We will make the simplifying assumption that the largest $\mu_i$ is unique and, without loss of generality, let us assume that the coordinates are permuted in order that $\mu_1$ is the largest mean reward. Furthermore, define $\Delta_i := \mu_1 - \mu_i$ for $i = 2, \ldots, K$.

A central piece of the analysis relies on the following potential function, which depends on the current number of plays of each arm $i = 2, \ldots, K$.

$$\Phi(N_2^t, \ldots, N_K^t) := 2 \sum_{i=2}^{K} \sum_{N=0}^{N_i^t - 1} \varsigma(N, \delta) \tag{2}$$

**Lemma 1.** *The expected regret of UCB is bounded as*

$$\mathbb{E}[\text{Regret}_T(\text{UCB})] \leq \mathbb{E}[\Phi(N_2^{T+1}, \ldots, N_K^{T+1})] + O(T\delta)$$

*Proof.* The (random) additional regret suffered on round $t$ of UCB is exactly $\mu_1 - \mu_{I^t}$. By virtue of our given deviation bound, we know that

$$\mu_1 \leq \hat{\mu}_1^t + \varsigma(N_1^t, \delta) \qquad \text{and} \qquad \hat{\mu}_{I^t}^t \leq \mu_{I^t} + \varsigma(N_{I^t}^t, \delta), \qquad \text{each w.p.} > 1 - \delta. \qquad (3)$$

Also, let $\xi^t$ be the indicator variable that one of the above two inequalities fails to hold. Of course we chose $\varsigma(\cdot)$ in order that $\mathbb{E}[\xi^t] \leq 2\delta$ via a simple union bound.

Note that, by virtue of using the UCB selection rule for $I^t$, it is clear that we have

$$\hat{\mu}_1^t + \varsigma(N_1^t, \delta) \leq \hat{\mu}_{I^t}^t + \varsigma(N_{I^t}^t, \delta) \qquad (4)$$

If we combine Equations 3 and 4, and consider the event that $\xi^t = 0$, then we obtain

$$\mu_1 \leq \hat{\mu}_1^t + \varsigma(N_1^t, \delta) \leq \hat{\mu}_{I^t}^t + \varsigma(N_{I^t}^t, \delta) \leq \mu_{I^t} + 2\varsigma(N_{I^t}^t, \delta).$$

Even in the event that $\xi^t = 1$ we have that $\mu_1 - \mu_{I^t} \leq 1$. Hence, it follows immediately that $\mu_1 - \mu_{I^t} \leq 2\varsigma(N_{I^t}^t, \delta) + \xi^t$.

Finally, we observe that the potential function was chosen so that $\Phi(N_2^{t+1}, \ldots, N_K^{t+1}) - \Phi(N_2^t, \ldots, N_K^t) = 2\varsigma(N_{I^t}^t, \delta)$. Recalling that $\Phi(0, \ldots, 0) = 0$, a simple telescoping argument gives that

$$\mathbb{E}[\text{Regret}_T(\text{UCB})] \leq \mathbb{E}\left[\Phi(N_2^{T+1}, \ldots, N_K^{T+1}) + \sum_{t=1}^{T} \xi^t\right] = \mathbb{E}[\Phi(N_2^{T+1}, \ldots, N_K^{T+1})] + 2T\delta.$$

$\square$

The final piece we need to establish is that the number of pulls $N_i^t$ of arm $i$, for $i = 2, \ldots, K$, is unlikely to exceed $\sharp(\Delta_i, \delta)$. This result uses some more standard techniques from the original UCB analysis (Auer et al., 2002), and we defer it to the appendix.

**Lemma 2.** *For any $T > 0$ we have* $\mathbb{E}[\Phi(N_2^{T+1}, \ldots, N_K^{T+1})] \leq \Phi(\sharp(\Delta_2, \delta), \ldots, \sharp(\Delta_K, \delta)) + O(T^2\delta)$.

We are now able to combine the above results for the final bound.

**Theorem 1.** *If we set $\delta = T^{-2}/2$, the expected regret of UCB is bounded as*

$$\mathbb{E}[\text{Regret}_T(\text{UCB})] \leq 8 \sum_{i=2}^{K} \frac{\log(T)}{\Delta_i} + O(1).$$

*Proof.* Note that a very standard deviation bound that holds for *all* distributions supported on $[0,1]$ is the *Hoeffding-Azuma* inequality (Cesa-Bianchi and Lugosi, 2006), where the bound is given by $f(N, \varepsilon) = 2\exp(-2N\varepsilon^2)$. Utilizing Hoeffding-Azuma we have $\sharp(\varepsilon, \delta) = \left\lceil \frac{2\log 2/\delta}{\varepsilon^2} \right\rceil$ and $\varsigma(N, \delta) = \sqrt{\frac{\log(2/\delta)}{2N}}$ for $N > 0$. If we utilize the fact that $\sum_{y=1}^{Y} \frac{1}{\sqrt{y}} \leq 2\sqrt{Y}$, then we see that

$$\Phi(\sharp(\Delta_2, \delta), \ldots, \sharp(\Delta_K, \delta)) = 2 \sum_{i=2}^{K} \sum_{N=0}^{\sharp(\Delta_i, \delta)} \varsigma(N, \delta) = 2 \sum_{i=2}^{K} 2\sqrt{\frac{\log(2/\delta)\sharp(\Delta_i, \delta)}{2}} = 4 \sum_{i=2}^{K} \frac{\log(2/\delta)}{\Delta_i}.$$

Combining the Lemma 1 and Lemma 2, setting $\delta = T^{-2}/2$, we conclude the theorem. $\square$

## 3 The Threshold Bandits Model

In the preceding, we described a potential-based proof for the UCB algorithm in the classic stochastic bandit problem. We now return to the Threshold Bandit setting, our problem of interest.

A $K$-armed Threshold Bandit problem is defined by random variables $X_i^t$ and a sequence of threshold values $c^t$ for $1 \leq i \leq K$ and $1 \leq t \leq T$, where $i$ is the index for arms. Successive pulling of arm $i$ generates the values $X_i^1, X_i^2, \ldots, X_i^T$, which are drawn i.i.d. from an unknown distribution. The threshold values $c^1, c^2, \ldots, c^T$ are drawn from $M = \{1, 2, \ldots, m\}$ (according to rules specified later). The threshold value $c^t$ is observed at the beginning of round $t$, and the learner follows a policy $P$ to choose the arm to play based on its past selections and previously observed feedbacks. Suppose the arm pulled at round $t$ is $I^t$, the observed reward is then $R_{I^t}^t = \mathbb{I}[X_{I^t}^t \geq c^t]$; that is, we receive a unit reward when the sample $X_{I^t}^t$ exceeds the threshold value $c^t$, and otherwise we receive no reward. We distinguish two different types of feedback.

1. **Uncensored Feedback:** After playing arm $I^t$, the learner observes the sample $X_{I^t}^t$ regardless of the threshold value $c^t$.

2. **Censored Feedback:** After playing $I^t$, the learner observes[2] $\begin{cases} \emptyset & \text{if } X_{I^t}^t \geq c^t, \\ X_{I^t}^t & \text{otherwise} \end{cases}$.

   In this case, we refer to the threshold value as a *censor value*.

Let $F_i(x)$ denote the survival function of the distribution on arm $i$. That is, $F_i(x) = \Pr(X_i^t \geq x)$. We measure regret against the optimal policy with full knowledge of $F_1, \ldots, F_n$ *i.e.*,

$$\text{Regret}_T(P) = \mathbb{E}\left[\sum_{t=1}^{T}\left(\max_{i \in [n]} R_i^t - R_{I^t}^t\right)\right] = \mathbb{E}\left[\sum_{t=1}^{T}\left(\max_{i \in [n]} \mathbb{I}\left(X_i^t \geq c^t\right) - \mathbb{I}\left(X_{I^t}^t \geq c^t\right)\right)\right].$$

Notice that for a fixed value of $c^t$, each arm $i$ has expected payoff $\mathbb{E}[R_i^t] = F_i(c^t)$, the regret can also be written as

$$\text{Regret}_T(P) = \mathbb{E}\left[\sum_{t=1}^{T}\left(\max_{i \in [n]} F_i(c^t) - F_{I^t}(c^t)\right)\right].$$

Our goal is to design a policy that minimizes the regret.

## 4    DKWUCB: Dvoretzky-Kiefer-Wolfowitz Inequality based Upper Confidence Bound algorithm

In this section, we study the uncensored feedback setting in which the value $X_{I^t}^t$ is always observed regardless of $c^t$. We assume that the largest $F_i(j)$ is unique for all $j \in M$, and define $i^*(j) = \arg\max_i F_i(j)$, $\Delta_i(j) = F_{i^*(j)}(j) - F_i(j)$ for all $i = 1, 2, \ldots, K$ and $j \in M$.

Under this setting, the algorithm will use the empirical distribution as an estimate for the true distribution. Specifically, we want to estimate the true survival function $F_i$ via:

$$\hat{F}_i^t(j) = \frac{\sum_{\tau=1}^{t-1} \mathbb{I}[X_{I^\tau}^\tau \geq j, I^\tau = i]}{N_i^t} \quad \forall j \in M \tag{5}$$

The key tool in our analysis is a deviation bound on the empirical CDF of a distribution, and we note that this bound holds uniformly over the support of the distribution. The Dvoretzky-Kiefer-Wolfowitz (DKW) inequality (Dvoretzky et al., 1956) allows us to bound the error on $\hat{F}_i^t(j)$:

**Lemma 3.** *At a time $t$, let $\hat{F}_i^t$ be the empirical distribution function of $F_i$ as given in equation 5. The probability that the maximum of the difference between $\hat{F}_i^t$ and $F_i$ over all $j \in M$ is at least $\varepsilon$ is less than $2\exp\left(-2\varepsilon^2 N_i^t\right)$, i.e.,*

$$\Pr\left(\sup_{j \in M} |\hat{F}_i^t(j) - F_i(j)| \geq \varepsilon \mid N_i^t \geq N\right) \leq 2\exp\left(-2\varepsilon^2 N\right).$$

The proof of the lemma can be found in Dvoretzky et al. (1956). The key insight is that the estimate $\hat{F}_i$ converges to $F_i$ point-wise at the same rate as the Hoeffding-Asumza inequality. That is, one does not pay an additional $M$ factor from applying a union bound. The fact that we have uniform convergence of the CDF with the same rate as the Hoeffding-Azuma inequality allows us to immediately apply the potential function argument from Section 2. In particular, we define $f(N, \varepsilon) = 2\exp\left(-2\varepsilon^2 N\right)$, as well as the pair of functions $\sharp(\varepsilon, \delta)$ and $\varsigma(N, \delta)$ exactly the same as the previous section, *i.e.*,

$$\sharp(\varepsilon, \delta) \quad := \quad \left\lceil \frac{2\log 2/\delta}{\varepsilon^2} \right\rceil,$$

$$\varsigma(N, \delta) \quad := \quad \begin{cases} \sqrt{\frac{\log(2/\delta)}{2N}} & \text{if } N > 0; \\ 1 & \text{otherwise.} \end{cases}$$

We are now ready to define our DKWUCB algorithm for a fixed choice of parameter $\delta > 0$ to solve the problem.

**DKWUCB Algorithm:**    on round $t$ play $I^t \leftarrow \arg\max_i \left\{\hat{F}_i^t(c^t) + \varsigma(N_i^t, \delta)\right\}$. $\tag{6}$

To analyze DKWUCB, we use a slight variant of the potential function defined in Section 2. Let $i^*(j) = \arg\max_i F_i(j)$ denote the optimal arm for threshold value $j$, and $\tilde{N}_i^t$ denote the number of rounds arm $i$ is pulled when it is not optimal, $\tilde{N}_i^t = \sum_{\tau=1}^{t-1} \mathbb{I}[I^\tau = i, I^\tau \neq i^*(c^\tau)]$. Notice that $\tilde{N}_i^t \leq N_i^t$. Define the potential function as:

$$\Phi(\tilde{N}_1^t, \ldots, \tilde{N}_K^t) := 2 \sum_{i=1}^{K} \sum_{N=0}^{\tilde{N}_i^t - 1} \varsigma(N, \delta) \tag{7}$$

**Theorem 2.** *Setting* $\delta = T^{-2}/2$, *the expected regret of DKWUCB is bounded as*

$$\mathbb{E}[\text{Regret}_T(\text{DKWUCB})] \leq 8 \sum_{i=1}^{K} \frac{\log T}{\min_{j \in M} \Delta_i(j)} + O(1),$$

We defer the proof of this theorem to the appendix.

We pause now to comment on some of the strengths of this type of analysis. At a high-level, the typical analysis to the UCB algorithm for the standard multi-armed bandit problem (Auer et al., 2002) is the following: (1) at some finite time $T$, the number of pulls of a bad arm $i$ is $O\left(\frac{\log(T)}{\Delta_i^2}\right)$ with high probability, and (2) the regret suffered by any such pull is $O(\Delta_i)$. The contribution of arm $i$ to total regret is therefore $O\left(\frac{\log(T)}{\Delta_i}\right)$. In contrast, we analyzed the UCB algorithm in Section 2 by observing that the expected regret suffered on round $t$ is bounded by the difference between the empirical mean estimator and the true mean for the payoff of arm $I^t$. Of course by design this quantity is almost certainly (w.p. at least $1 - \delta$) less than $\varsigma(N_{I^t}^t)$. The potential function $\Phi(\cdot, \ldots, \cdot)$ tracks the accumulation of these values $\varsigma(N_i^t)$ for each arm $i$, and the final regret bound is a consequence of the summation properties of $\varsigma, \sharp$ for the particular estimator being used.

While these two approaches lead to the same bound in the standard multi-armed bandit problem, the potential function approach bears fruit in the Threshold Bandit setting. Because the uniform convergence rate promised by the DKW inequality matches that of the Hoeffding-Azume inequality, Theorem 2 should not be surprising; the $i$th arm's contribution to DKWUCB's regret should be idenitical to UCB, but with the suboptimality gap now equal to $\min_j \Delta_i(j)$.

However, following the program for the standard analysis of UCB, one would naively argue that arm $i$ is incorrectly pulled $O\left(\frac{\log(T)}{(\min_{j \in M} \Delta_i(j))^2}\right)$ times. These pulls might come in the face of any number of threshold values $c^t$, suffering as much as $\max_{j \in M} \Delta_i(j)$ regret, yielding a bound of $O\left(\frac{\max_{j \in M} \Delta_i(j) \log(T)}{(\min_{j \in M} \Delta_i(j))^2}\right)$ on the $i$th arm's regret contribution, which is a factor $O\left(\frac{\max_j \Delta_i(j)}{\min_j \Delta_i(j)}\right)$ worse than the derived result. By tracking the convergence of the underlying estimator, we circumvent this problem entirely.

## 5 KMUCB: Kaplan-Meier based Upper Confidence Bound Algorithm

We now turn to the censored feedback setting, in which the feedback of pulling arm $I^t$ is observed only when $X_{I^t}^t$ is less than $c^t$. For ease of presentation, we assume that the largest $F_i(j)$ is unique for all $j \in M$, and define $i^*(j) = \arg\max_i F_i(j), \Delta_i(j) = F_{i^*(j)}(j) - F_i(j)$ for all $i = 1, 2, \ldots, K$ and $j \in M$.

One prevalent non-parametric estimator for censored data is the Kaplan-Meier maximum likelihood estimator Kaplan and Meier (1958); Peterson (1983). Most of existing works have studied the uniform error bound of Kaplan-Meier estimator in the case that the threshold values are drawn i.i.d. from a known distribution Foldes and Rejto (1981) or asymptotic error bound for the non-i.i.d. case Huh et al. (2009). The only known uniform error bound of Kaplan-Meier estimator is proposed in Ganchev et al. (2010).

Noting that for a given threshold value, all the feedbacks from larger threshold values are useful, we propose a new estimator with tighter uniform error bound based on the Kaplan-Meier estimator as following:

$$\hat{F}_i^t = \frac{D_i^t(j)}{N_i^t(j)} \tag{8}$$

where $D_i^t(j)$ and $N_i^t(j)$ is defined as follows

$$A^t := \min\{X_{I^t}^t, c^t\}, \qquad D_i^t(j) := \sum_{\tau=1}^{t-1} \mathbb{I}[A^\tau \geq j, I^\tau = i], \qquad N_i^t(j) := \sum_{\tau=1}^{t-1} \mathbb{I}[c^\tau \geq j, I^\tau = i].$$

We first present an error bound for the modified Kaplan-Meier estimate of $F_i(j)$ :

**Lemma 4.** *At time $t$, let $\hat{F}_i^t$ be the modified Kaplan-Meier estimate of $F_i$ as given in equation 8. For any $j \in M$, the probability that the difference between $\hat{F}_i^t(j)$ and $F_i(j)$ is at least $\varepsilon$ is less than* $2\exp\left(-\frac{\varepsilon^2 N_i^t(j)}{2}\right)$ *, i.e.,*

$$\Pr\left(|\hat{F}_i^t(j) - F_i(j)| \geq \varepsilon\right) \leq 2\exp\left(-\frac{\varepsilon^2 N_i^t(j)}{2}\right).$$

We defer the proof of this lemma to the appendix.

Different to the stochastic uncensored MAB setting, we show that the cost of learning with censored feedback depends significantly on the order of the threshold values. To illustrate this point, we first show a comparison between the regret of adversarial setting and optimistic setting. In the adversarial setting, the threshold values are chosen to arrive in a non-decreasing order $1, 1, \ldots, 1, 2, \ldots, 2, 3, \ldots, m$, the problem becomes playing $m$ independent copies of bandits, and the regret scales with $m$; while in the optimistic setting, the threshold values are chosen to arrive in a non-increasing order $m, m, \ldots, m, m-1, \ldots, m-1, \ldots, 1, \ldots, 1$, which means the learner can make full use of the samples, and can thus perform significantly better. Afterwards, we show that if the order of the threshold values is close to uniformly random, the regret only scales with $\log m$.

## 5.1 Adversarial vs. Optimistic Setting

For the simplicity of presentation, we assume that in both settings, the time horizon could be divided in to $m$ stages, each with length $\lfloor T/m \rfloor$.. In the adversarial setting, threshold value $j$ comes during stage $j$; while in the optimistic setting threshold value $m - j + 1$ comes during stage $j$.

For the adversarial setting, due to the censored feedback structure, only the samples observed within the same stage can help to inform decision making. From the perspective of the learner, this is equivalent to facing $m$ independent copies of stochastic MAB problems, and thus, the regret scales with $m$. Making use of the lower bound of stochastic MAB problems Lai and Robbins (1985), we can conclude the following theorem.

**Theorem 3.** *If the threshold values arrive according to the adversarial order specified above, no learning algorithm can achive a regret bound better than $\sum_{j=1}^{m} \sum_{i=1}^{K} \frac{\log(T/m)}{KL(B(F_i(j)||B(F_{i^*}(j)(j)))}$, where $KL(\cdot||\cdot)$ is the Kullback-Leibler divergence Lai and Robbins (1985) and $B(\cdot)$ is the probability distribution function of Bernoulli distribution.*

For the optimistic setting, although the feedbacks are right censored, we note that every sample observed in the previous rounds are useful in later rounds. This is because the threshold values arrive in non-increasing order. Therefore, we can reduce the optimistic setting to the Threshold Bandit problem with uncensored feedback, and use the DKWUCB proposed in Section 4 to solve it. More specifically, we can set

$$
\begin{aligned}
f(N, \varepsilon) &:= 2\exp(-\varepsilon^2 N/2), \\
\sharp(\varepsilon, \delta) &:= \left\lceil \frac{8\log 2/\delta}{\varepsilon^2} \right\rceil, \\
\varsigma(N, \delta) &:= \begin{cases} \sqrt{\frac{2\log(2/\delta)}{N}} & \text{if } N \geq 1; \\ 1 & \text{otherwise.} \end{cases}
\end{aligned}
$$

and on every round, the learner plays the same strategy as DKWUCB. We call this strategy OPTIM. Following the same procedure in Section 4, we can provide a regret for OPTIM.

**Theorem 4.** *Let $\delta = T^{-2}/2$ and assume $T \geq mK$. The regret of the optimistic setting satisfies*

$$\mathbb{E}[\text{Regret}_T(\text{OPTIM})] \leq 32 \sum_{i=1}^{K} \frac{\log T}{\min_{j \in M} \Delta_i(j)} + O(1).$$

## 5.2 Cyclic Permutation Setting

In this subsection, we show that if the order of threshold values is close to uniformly random, we can perform significantly better than the adversarial setting. To be precise, we assume that the threshold values are a cyclic permutation order of $1, 2, \ldots, m$. We define the set $M = \{c^{km}, c^{km+1}, \ldots, c^{k(m+1)-1}\}$ for any non-negative integer $k \leq T/m$.

We are now ready to present KMUCB, which is a modified Kaplan-Meier-based UCB algorithm. KMUCB divides the time horizon into epochs of length $Km$ and, for each epoch, pulls each arm once for each threshold value. KMUCB then performs an "arm elimination" process, and once all but one arm has been eliminated, it proceeds to pull the single remaining arm for the given threshold value. KMUCB's estimation procedure leverages information across threshold values, where observations from higher thresholds are utilized to estimate mean payoffs for lower thresholds; information does not flow in the other direction, however, as a result of the censoring assumption. Specifically, for a given threshold index $j$, KMUCB tracks the arm elimination process as follows: for any threshold values below $j$, KMUCB believes that we have determined the best arm, and plays that arm constantly. For threshold values greater than or equal to $j$, KMUCB explores all arms uniformly. Note that by uniform exploration over all arms for threshold value $j$, all sub-optimal arms can be detected with probability at least $1 - O\left(\frac{1}{T}\right)$ after $O\left(\frac{\log T}{(m-j+1)\min_{i \in [K]} \Delta_i^2(j)}\right)$ epochs. KMUCB then removes all the sub-optimal arms for threshold value $j$, and increments $j$ by 1. Denoting the last time unit of epoch $k$ as $t_k = kKm$, the detailed description of KMUCB is shown in Algorithm 1.

---

**Algorithm 1** KMUCB

---

1: **Input:** A set of arms $1, 2, \ldots, K$.
2: **Initialization:** $L_j \leftarrow [K] \; \forall j \in M, k \leftarrow 1, j \leftarrow 1$
3: **for** epoch $k = 1, 2, \ldots, T/Km$ **do**
4:      count$[j'] \leftarrow 0 \; \forall j' \in M$
5:      **for** $t$ from $(t_{k-1} + 1)$ to $t_k$ **do**
6:          **Observe** $c^t = j'$ and set count$[j'] \leftarrow$ count$[j'] + 1$
7:          **if** $j' < j$ **then**
8:              $I^t \leftarrow$ index of the single arm remaining in $L_{j'}$
9:          **else**
10:             $I^t \leftarrow$ count$[j']$.
11:          **end if**
12:      **end for**
13:      **if** $j \leq m$ **and** $\max_{i' \in [K]} \hat{F}_{i'}^{t_k}(j) - \hat{F}_i^{t_k}(j) \geq \sqrt{\frac{16 \log(Tk)}{(m-j+1)k}} \; \forall i \in L_j \setminus \{\arg\max_{i' \in [K]} \hat{F}_{i'}^{t_k}(j)\}$ **then**
14:

$$L_j \leftarrow \left\{ \arg\max_{i' \in [K]} \hat{F}_{i'}^{t_k}(j) \right\}, \qquad j \leftarrow j + 1$$

15:      **end if**
16: **end for**

---

**Theorem 5.** *The expected regret of KMUCB is bounded as*

$$\sum_{i=1}^{K} \log m \frac{128 \max_{j \in M} \Delta_i(j) \log T}{\min_{i \in [K], j \in M} \Delta_i^2(j)} + O(1).$$

We defer the proof of this theorem to the appendix.

We note two directions of future research. First, we believe the above bound can likely be made stronger by either improving upon the minimization in the denominator or the maximization in the numerator. Second, we believe the "cyclic permutation" assumption can be weakened to "uniformly randomly sequence of thresholds," but we were unable to make progress in this direction. We welcome further investigation along these lines.

## Footnotes

[1]Note that in much of our notation we use superscript $t$ to denote the time period rather than as an exponent.

[2]Existing literature often refers to this as right-censoring. With right-censored feedback, samples from playing arms at high threshold values can inform decisions at low threshold values but not vice versa.

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
