[Supplementary Material · nips_2016_appendix.pdf]

# A Omitted Proofs

*Proof of Lemma 2.* To obtain the inequality of the lemma, define for every $t = 1, \ldots, T$ and $i = 2, \ldots, K$ the indicator variable $\zeta_i^t$ which returns 1 when $I^t = i$ *and* $N_i^t \geq \sharp(\Delta_i, \delta)$, and returns 0 otherwise. We can show that $\zeta_i^t = 1$ with probability smaller than $2\delta$.

Note that if $I^t = i$ then the upper confidence estimate for $i$ was larger than that of action 1. More precisely, it must be that $\hat{\mu}_i^t + \varsigma(N_i^t) \geq \hat{\mu}_1^t + \varsigma(N_1^t)$. For this to occur, either we had (a) a large underestimate on $\mu_1$, that is $\hat{\mu}_1^t + \varsigma(N_1^t) \leq \mu_1$. Or, (b) we had a major overestimate on $\mu_i$, that is, $\hat{\mu}_i^t + \varsigma(N_i^t) \geq \mu_1$. It is clear that (a) occurs with probability less than $\delta$ by construction of $\varsigma$.

To analyze (b), note that $\mu_1 = \mu_i + \Delta_i$, and we are also given that $N_i^t \geq \sharp(\Delta_i, \delta)$ which implies that $\varsigma(N_i^t) \leq \Delta_i/2$.

$$\hat{\mu}_i^t + \varsigma(N_i^t) \geq \mu_1 \implies \hat{\mu}_i^t \geq \mu_i + \varsigma(N_i^t) \implies \hat{\mu}_i^t - \mu_i \geq \varsigma(N_i^t),$$

and of course the latter happens with probability no more than $\delta$.

Since

$$\Phi(N_2^{T+1}, \ldots, N_K^{T+1}) = 2\sum_{i=1}^{K} \sum_{N=0}^{N_i^{T+1}} \varsigma(N) \leq 2\sum_{i=1}^{K} \left( \sum_{N=0}^{\sharp(\Delta_i, \delta)} \varsigma(N) + \sum_{t=1}^{T} \zeta_i^t \right)$$

We can conclude that

$$\mathbb{E}[\Phi(N_2^{T+1}, \ldots, N_K^{T+1})] \leq \Phi(\sharp(\Delta_2, \delta), \ldots, \sharp(\Delta_K, \delta)) + 2\mathbb{E}[\sum_{i=2}^{K} \sum_{t=1}^{T} \zeta_i^t] \leq \Phi(\sharp(\Delta_2, \delta), \ldots, \sharp(\Delta_K, \delta)) + 4T^2\delta.$$

$\square$

*Proof Sketch of Theorem 2.* The proof follows much in the same way as that of Theorem 1. The regret suffered on round $t$ of DKWUCB is exactly $F_{i^*(c^t)}(c^t) - F_{I^t}(c^t)$. Let $\xi^t$ be the indicator equal to 0 on the event that both $F_{i^*(c^t)}(c^t) \leq \hat{F}_{i^*(c^t)}(c^t) + \varsigma(N_{I^t}^t, \delta)$ and $\hat{F}_{I^t}(c^t) \leq F_{I^t}(c^t) + \varsigma(N_{I^t}^t, \delta)$, where $\varsigma$ was chosen so that $E[\xi^t] \leq 2\delta$. On $\xi^t = 0$, the KWUCB selection rule guarantees that $F_{i^*(c^t)}(c^t) - F_{I^t}(c^t) \leq 2\varsigma(N_{I^t}^t, \delta) \leq 2\varsigma(\tilde{N}_{I^t}^t, \delta)$ (c.f. Lemma 1 for details). Thus, we can bound $F_{i^*(c^t)}(c^t) - F_{I^t}(c^t) \leq 2\varsigma(\tilde{N}_{I^t}^t, \delta) + \xi^t$.

Noting that $\Phi(\tilde{N}_1^{t+1}, \ldots, \tilde{N}_K^{t+1}) - \Phi(\tilde{N}_1^t, \ldots, \tilde{N}_K^t) = 2\varsigma(\tilde{N}_{I^t}^t, \delta)$ when $i^*(c^t) \neq I^t$ and 0 otherwise, we can bound cummulative regret by telescoping $\Phi$, giving us

$$\mathbb{E}[\text{Regret}_T(\text{DKWUCB})] \leq \mathbb{E}[\Phi(\tilde{N}_1^{T+1}, \ldots, \tilde{N}_K^{T+1})] + \mathbb{E}[\sum_{t=1}^{T} \xi^t] \leq \mathbb{E}[\Phi(\tilde{N}_1^{T+1}, \ldots, \tilde{N}_K^{T+1})] + 2T\delta$$

Now define $\zeta_i^t$ as the indicator variable which returns one on the event that $I^t = i$, $i^*(c^t) \neq i$, and $\tilde{N}_i^t > \min_{j \in M} \Delta_i(j)$. By similar arguments to Lemma 2, one can bound $\mathbb{E}[\zeta_i^t] \leq 2\delta$. Since with probability one $\Phi(\tilde{N}_1^{T+1}, \ldots, \tilde{N}_K^{T+1}) \leq 2\sum_{i=1}^{K} \left( \sum_{N=0}^{\sharp(\min_j \Delta_i(j), \delta)} \varsigma(N) + \sum_{t=1}^{T} \zeta_i^t \right)$, we have

$$\mathbb{E}[\text{Regret}_T(\text{DKWUCB})] \leq \Phi(\sharp(\min_j \Delta_1(j), \delta), \ldots, \sharp(\min_j \Delta_K(j), \delta)) + O(T^2\delta)$$

The remainder of the proof follows identically to that of Theorem 1, by bounding the sum $\Phi(\sharp(\min_j \Delta_1(j), \delta), \ldots, \sharp(\min_j \Delta_K(j), \delta)) = \sum_{i=1}^{K} \sum_{N=0}^{\sharp(\min_j \Delta_i(j), \delta)} \varsigma(N, \delta)$, and tuning $\delta$.

$\square$

*Proof of Lemma 4.* At time $t$, let $t_k$ be the index $\tau$ of the $k$th time step at which $I^\tau = i$ and $c^\tau \geq j$. By defenition, we have that $k \leq N_i^t(j)$. We also define $Y_0 = 0$ and $Y_k = \sum_{l=1}^{k} (F_i(j) - \mathbb{I}[A^{t_l} \geq j])$ for each

$k \in \{0, 1, 2, \ldots, N_i^t(j)\}$. Note that

$$
\begin{aligned}
\mathbb{E}\left[Y_k | Y_{k-1}, \ldots, Y_1\right] &= \mathbb{E}[Y_k | Y_{k-1}] = \mathbb{E}\left[Y_{k-1} + F_i(j) - \mathbb{I}[A^{t_k} \geq j] | Y_{k-1}\right] \\
&= \mathbb{E}[Y_{k-1} | Y_{k-1}] + \mathbb{E}\left[F_i(j) - \mathbb{I}[A^{t_k} \geq j] | Y_{k-1}\right] \\
&= Y_{k-1} + \mathbb{E}\left[F_i(j) - \Pr(X_i^{t_k} \geq j | c^{t_k} \geq j) | Y_{k-1}\right] \\
&= Y_{k-1} + \mathbb{E}\left[F_i(j) - F_i(j) | Y_{k-1}\right] = Y_{k-1}.
\end{aligned}
$$

Therefore, the sequence $Y_1, Y_2, \ldots, Y_{N_i^t(j)}$ forms a martigale, and $|Y_k - Y_{k-1}| \leq 1$ for each $k$. By Azuma's inequality, for any $\varepsilon > 0$,

$$
\Pr\left(|Y_{N_i^t(j)}| \geq \varepsilon N_i^t(j)\right) \leq 2\exp\left(-\frac{\varepsilon^2 N_i^t(j)}{2}\right).
$$

Note that $Y_{N_i^t(j)} = N_i^t(j)\left(F_i(j) - \hat{F}_i^t(j)\right)$, we have

$$
\Pr\left(|\hat{F}_i^t(j) - F_i(j)| \geq \varepsilon\right) \leq 2\exp\left(-\frac{\varepsilon^2 N_i^t(j)}{2}\right),
$$

which concludes the lemma. $\qquad\square$

*Proof of Theorem 5.* At an epoch $k$, denote the set of arm indices at the beginning as $L_j^k$ for a threshold value $j \in M$. By the property of uniform exploration, we have

$$
N_i^{t_k}(j) \geq (m - j + 1)k \qquad \forall i \in L_j^k. \tag{9}
$$

By Lemma 4, we have that for any $i \in L_j^k$, the difference between $F_i(j)$ and $\hat{F}_i^{t_k}(j)$ is upper bounded by $\sqrt{4\log(Tk)/(m-j+1)k}$ with probability at least $1 - 1/T^2k^2$, *i.e.*,

$$
\Pr\left(|\hat{F}_i^{t_k}(j) - F_i(j)| \geq \sqrt{\frac{4\log(Tk)}{(m-j+1)k}}\right) \leq \frac{1}{T^2k^2}. \tag{10}
$$

Note that $\sum_{k=1}^{\infty} 1/k^2 = \pi^2/6 < 2$, by applying union bound three times, we have

$$
\Pr\left(|\hat{F}_i^{t_k}(j) - F_i(j))| \leq \sqrt{\frac{4\log(Tk)}{(m-j+1)k}}\right) \geq 1 - \frac{2}{T} \tag{11}
$$

holds for all epoch index $k$, arm index $i$, and threshold value $j$. Therefore, with probability at least $1 - 2/T$, $i^*(j)$ is never eliminated from $L_j$ for all $j \in M$. Therefore, the expected regret for missed elimination is $O(T \cdot 2/T) = O(1)$.

We then bound the number of times a sub-optimal arm is pulled for a level $j$ conditioning on $i^*(j)$ is not eliminated from $L_j^k$ for $k = 1, 2, \ldots, T/Km$. In the worst case, to eliminate all sub-optimal arms $i$ from $L_j^k$, KMUCB needs to come to an epoch $k$ such that

$$
|L_{j'}| = 1 \ \forall j' < j, \tag{12}
$$

$$
\hat{F}_{i^*(j)}^{t_k}(j) - \hat{F}_i^{t_k}(j) \geq \sqrt{\frac{16\log(Tk)}{(m-j+1)k}} \ \forall i \in [K] \setminus \{i^*(j)\}. \tag{13}
$$

By Equation 11, we have with probability at least $1 - 4/T$,

$$
\hat{F}_{i^*(j)}^{t_k}(j) \geq F_{i^*(j)}(j) - \sqrt{\frac{4\log(Tk)}{(m-j+1)k}} \tag{14}
$$

and

$$
\hat{F}_i^{t_k}(j) \leq F_i(j) + \sqrt{\frac{4\log(Tk)}{(m-j+1)k}} \ \forall i \in [K] \tag{15}
$$

hold. Therefore, if for all $i \neq i^*(j)$, $k$ satisfies that

$$F_{i^*(j)}(j) - \sqrt{\frac{4\log(Tk)}{(m-j+1)k}} - \left( F_i(j) + \sqrt{\frac{4\log(Tk)}{(m-j+1)k}} \right) \geq \sqrt{\frac{16\log(Tk)}{(m-j+1)k}}, \quad (16)$$

which is equivalen to

$$k \geq \frac{128\log T}{(m-j+1)\min_{i\in[K]}\Delta_i^2(j)}, \quad (17)$$

then with probability at least $1 - 4/T$, $k$ satisfies inequality 13.

But ineuality (12) implies that $k$ would also need to be greater than or equal to $\max_{j'<j}\frac{128\log T}{(m-j+1)\min_{i\in[K]}\Delta_i^2(j)}$ in order all for sub-optimal arms to be removed from $L_j^k$. Therefore, it would be sufficient to set

$$k = \frac{128\log T}{(m-j+1)\min_{i\in[K],j\in M}\Delta_i^2(j)}. \quad (18)$$

Using the fac that $\sum_{j=1}^m \frac{1}{j} = \log m$, this contributes a total of $\sum_{i=1}^K \log m \frac{128\max_{j\in M}\Delta_i(j)\log T}{\min_{i\in[K],j\in M}\Delta_i^2(j)}$ to the regret. Also noting that the regret comes from failing to remove any sub-optimal index $i$ from any $L_j$ is $O(T \cdot 4/T) = O(1)$, the total regret is

$$\sum_{i=1}^K \log m \frac{128\max_{j\in M}\Delta_i(j)\log T}{\min_{i\in[K],j\in M}\Delta_i^2(j)} + O(1). \quad (19)$$

$\square$