[Reviews · NeurIPS 2016]

Reviewer 1

Summary

The paper at hand studies multiarmed bandit problems with censored or uncensored feedback. In the both cases, one is given a threshold in each iteration and the goal is to choose between various random variables to maximize the chance of obtaining a value below the threshold. The difference between censored and uncensored is which information can be observed. For both cases, the paper presents algorithms and derives bounds for the expected regret.

Qualitative Assessment

I do not know much about the area and cannot really judge the relevance of this paper in the context of previous work. What I can say is that the problem is well introduced and well motivated and looks interesting to me. Furthermore, the technical novelty is well described and explained.

Confidence in this Review

1-Less confident (might not have understood significant parts)


Reviewer 2

Summary

The paper introduces a novel type of stochastic multi-armed bandit problem in which the objective when choosing an arm at time t is to maximize the probability of obtaining a reward greater than a threshold value specified as side information at time t. Two feedback models are considered: uncensored feedback, where the reward of the chosen arm is observed, and censored feedback, where the reward is only observed if it falls below the threshold. The paper presents upper-confidence-bound based algorithms for both cases, with regret guarantees that improve upon the naive bounds one would obtain by "decoupling" the learning problems associated to different threshold values. A key technical tool underpinning the proofs is a novel way of analyzing UCB algorithms via a potential function.

Qualitative Assessment

Technical quality: this a theory paper with no experimental results. I didn't check every step of the proofs, but those that I checked looked sound. I find the application of Dvoretsky-Kiefer-Wolfowitz and Kaplan-Meier in this context appealing, and the new potential function argument for analyzing UCB algorithms may be applicable elsewhere. Novelty/originality: the problem introduced in the paper is novel, although there is some similarity to the "dark pools" problem introduced by Goncharev et al. (UAI 2009), which the authors cite. The algorithms used to solve the problem are standard UCB algorithms into which the authors have plugged novel estimators based on the novel feedback structure in their problem. (I view this as a positive feature: UCB is a powerful technique and it's desirable to apply it to new problems rather than "reinventing the wheel".) The use of the Kaplan-Meier estimator in Section 5 is prefigured by the Goncharev et al. paper. Despite this superficial similarity, the problems that the two papers address are quite different. Potential impact: I think the potential impact is mainly theoretical rather than practical. I think the paper introduces a nice new wrinkle into the standard multi-armed bandit model and that the potential-function-based analysis of UCB is a nice innovation that will potentially find its way into other theory papers on bandits, but my high evaluation of paper is based more on elegance than on impact. Clarity and presentation: There are spelling errors sprinkled throughout the paper, e.g. "Wolfowit", "simpy", "preceeding". Other than that, I thought the clarity was admirable.

Confidence in this Review

2-Confident (read it all; understood it all reasonably well)


Reviewer 3

Summary

The following version of a stochastic MAB problem is studied. At any round t the learner is shown a real number c^t . If the learner chooses to pull arm i he observes X_i (uncensored case) or X_i 1{X_i < c^t} (censored case) and receives a binary reward R^t_i with Pr(R^t_i =1)=F_i(c^t), where F_i is an unknown. The paper presents a very brief discussion the classical MAB with outcomes that have a distribution with support in [0.1] and derive an asymptotic finite horizon bound for the expected regret, for a proposed UCB type of algorithm (Theorem 1). For the uncensored case, it presents a UCB type algorithm for which asymptotic finite horizon bounds of the expected regret are give in Theorem 2. For the censored case the presentation in incomplete and it provides partial results, e.g., Theorems 3 4 5 for different models of the c^t .

Qualitative Assessment

The model is interesting however the work from section 4 to the end seems to have been written in a harry and is very hard to follow. I am concerned that editorial comments of the authors can be misleading, i.e., the reader may think that the asymptotic finite horizon bounds they present are stronger results than the asymptotic results of Lai Robbins (85). In addition the authors seem to ignore important literature: Katehakis and Robbins (95) and Burnetas and Katehakis (1996) that provided the first `pure' UCBs, for Normal populations (with known variances) and for multi-parameter distributions. All these 3 papers provide asymptotic results with optimal constants. I believe that the constants given in this paper are far from being optimal in any sense.

Confidence in this Review

3-Expert (read the paper in detail, know the area, quite certain of my opinion)


Reviewer 4

Summary

The main focus of the paper is on the threshold bandit model in censored and uncensored feedback setting and the fresh perspective to prove classical UCB algorithm by potential function argument. The novel contribution of the paper is found in the implementation of Dvoretzky-Kiefer-Wolfowitz inequality to prove that the regret bound of uncensored feedback is no worse than stochastic MAB setting and the modified Kaplan Meier estimator based algorithm for censored feedback setting in optimistic, adversarial and cyclic permutation order of the threshold value.

Qualitative Assessment

The result and the theoretical contribution to the paper looks very promising. The use of potential function argument to prove the classical UCB algorithm is interesting and the result on the uncensored feedback and analysis is very promising. The meaning of the variable used in the numerator and denominator in the Kaplan Meier estimator in section 5 is not clear to me. Overall the paper is well written except for a few typos. The measure of regret has been done against optimal policy instead of the best arm. It is a bit unusual. The problem is important and the presented results are well established and correct to the best of my knowledge. A bit of experimental result supporting the theoretical result would be nice.

Confidence in this Review

1-Less confident (might not have understood significant parts)


Reviewer 5

Summary

The authors begin by providing an alternative proof for the regret bound of UCB. The endeavor of this seems to be to introduce the "potential trick" which the authors later use in the threshold setting. To my understanding, the threshold setting considered in this paper is a generalization of the UCB setting provided that the reward functions are all Bernoulli 0-1. To see this, one can fix c^t=c for all t. Then clearly the dominating arm is the one that maximizes $P(X^t_i \ge c)$. The distribution of the reward function $D_i$ is $1 w.p. P(X^t_i \ge c)$ and 0 w.p. 1- $P(X^t_i \ge c)$. I was wondering if a more appropriate generalization would be to reward $X^t_j$ provided it crosses the threshold. Probably in the setting of Dark pool brokerage this setting may find an application. Coming back to the threshold UCB described in the paper, supposing $c^t=c$ for all $t$ then we are back in the UCB setting. When the sequence of thresholds is predetermined, one does not expect the analysis to deviate too much from Auer et. al. Having said that, as the authors state, the potential trick certainly fits in well. The non-trivial threshold setting (uncensored and censored) is when $c^t$ is non-deterministic. Provided that $c^t$ takes values from a finite set the authors provide a regret bound that uses the DKW inequality to determine $f$, which in turn is used to fix the two auxiliary functions that allows for the potential trick to be applied. As the authors explain, one cannot just count the number of bad-arm-pulls especially when $c^t$ is not constant. For the setting of the censored feedback setting, the authors provide a new estimator based on the Kaplan-Meier estimator, then go on to find $f$ (error bound for the estimator). The regret bounds should now proceed as in the previous sections. Finally, they illustrate that the ordering of the threshold values affect the cost of learning. This is to be expected. The cost also seems to depend on how many values the threshold can take.

Qualitative Assessment

It would be interesting if the authors can justify why are the new rewards are 0-1 and not $X^t_j$ or in general some different distribution provided the threshold conditions are met. It seems that the likeness of UCB and DKWUCB is due to the uniform convergence of the empirical distributions to their "means". Does this mean that we can let $c^t$ take values over a continuous range provided such uniform-convergence-property hold. There should be a way to get around assuming that argmax_i F_i(j) is unique. This would certainly make it more applicable. Or is it that the $c^t$ may be perturbed to justify the assumption. It would be interesting if the authors can provide a detailed setting in a practical application wherein the $c^t$'s are non-deterministic/uniformly distributed over a finite set. The potential trick seems to capture the scenario in sequential decision settings wherein the "good arm"/right action changes with time. Can there be more insight into where the potential trick will prove helpful?

Confidence in this Review

2-Confident (read it all; understood it all reasonably well)


Reviewer 6

Summary

The paper consider a novel MAB setting called threshold MAB, where a threshold value is used to determine the reward of each round. In such a case, the authors further consider two type of feedback: uncensored and censored feedbacks where the main difference is whether the learner can view the sample returned by an arm.

Qualitative Assessment

I think the paper justify the threshold bandit very well by using the examples of delivery and supplier selection. The results are also promising that the uncensored threshold bandit is no harder than usual MAB problem.

Confidence in this Review

1-Less confident (might not have understood significant parts)